# Bacteriophages, Antibiotics and Probiotics: Exploring the Microbial Battlefield of Colorectal Cancer

**DOI:** 10.3390/ijms26167837

**Published:** 2025-08-13

**Authors:** Cristian Constantin Volovat, Mihai Andrei Cosovanu, Madalina-Raluca Ostafe, Iolanda Georgiana Augustin, Constantin Volovat, Bogdan Georgescu, Simona Ruxandra Volovat

**Affiliations:** 1Department of Radiology, “Grigore T. Popa” University of Medicine and Pharmacy, 700115 Iasi, Romania; cristian.volovat@yahoo.com; 2Center of Oncology Euroclinic, 700106 Iasi, Romania; 3Department of Medical Oncology-Radiotherapy, “Grigore T. Popa” University of Medicine and Pharmacy, 16 University Street, 700115 Iasi, Romania; madalina.ostafe@gmail.com (M.-R.O.); volovat.constantin@umfiasi.ro (C.V.); simonavolovat@gmail.com (S.R.V.); 4Department of Medical Oncology, Al. Trestioreanu Institute of Oncology, 022328 Bucharest, Romania; 5Department of Oncology, “Carol Davila” University of Medicine and Pharmacy, 050474 Bucharest, Romania

**Keywords:** colorectal cancer, dysbiosis, intestinal microbiome, bacteriophages, antibiotics, probiotics

## Abstract

Colorectal cancer (CRC), a prevalent malignancy, is a significant global health concern. The intricate interplay of genetic mutations, inflammatory processes, and environmental factors underscores the complexity of CRC’s etiology. The human gut harbors a diverse microbial community that plays a key role in maintaining homeostasis and influencing various aspects of host physiology. Perturbations in the gut microbiome (GM) composition and function have been implicated in CRC carcinogenesis. This bidirectional relationship involves microbial contributions to inflammation, DNA damage, and immune modulation, shaping the tumor microenvironment (TME). Bacteriophages, viruses that infect bacteria, contribute to the microbiome’s diversity and function by influencing bacterial abundance and composition. These phages can impact host–microbiome interactions, potentially influencing CRC risk. Furthermore, they can be manipulated to transport targeted medication, without being metabolized. Antibiotics exert selective pressures on the gut microbiome, leading to shifts in bacterial populations and potential dysbiosis. Probiotics can modulate the composition and activity of the GM and could be considered adjunctive therapy in the treatment of CRC. Understanding the intricate balance between bacteriophages, antibiotics–probiotics, and the GM is essential for comprehending CRC etiology and progression.

## 1. Introduction

Colorectal cancer (CRC) is the third most commonly diagnosed cancer worldwide and ranks as the second leading cause of cancer-related mortality. The global burden of CRC is to rise, with an estimated 3.3 million new cases expected by 2040. CRC develops through a multistep process beginning with benign polyps, progressing to adenomas, and culminating in malignant tumors. This transformation is driven by a complex interplay of genetic and environmental factors and often occurs over many years [1,2,3].

Among environmental influences, the gut microbiome (GM) has gained increasing attention for its role in CRC pathogenesis. Patients with CRC exhibit distinct gut microbial compositions compared to healthy individuals. Often described as the “forgotten organ,” the commensal gut microbiota plays a crucial role in maintaining host health by modulating various physiological processes. Emerging evidence supports a causal link between intestinal microbial dysbiosis and CRC development [4,5,6].

The gut bacteria residing primarily in the large intestine engage in constant interactions with colonic epithelial cells and other microbes. These interactions regulate key host functions, including energy metabolism and immune responses [2]. Furthermore, the GM and its metabolites can induce epigenetic modifications in host cells, serving as critical mediators in the complex crosstalk between microbiota and host [7]. A balanced GM promotes anti-cancer effects partly by producing beneficial metabolites such as short-chain fatty acids (SCFAs), which possess antioxidant and anti-inflammatory properties, strengthen the intestinal barrier, and provide energy substrates. In contrast, the altered GM in CRC patients may contribute directly to tumorigenesis [8].

Recently, research investigating the link between GM alterations and gastrointestinal disorders, particularly CRC, has expanded significantly. The distinct microbial shifts observed in CRC patients suggest that host–microbe interactions may play a pivotal role in cancer initiation and progression, offering novel strategies for prevention, early detection, and treatment [9].

An important area of cancer research focuses on understanding the variability in treatment response among patients with otherwise similar clinical profiles. The patient’s microbiome has emerged as a potential key factor influencing the efficacy of systemic cancer therapies [10].

This narrative review aims to synthesize current knowledge on the role of the gut microbiome in CRC development and treatment (Figure 1), highlighting the potential implications for future clinical practice and research.

## 2. Gut Microbiota

The human body harbors an enormous and complex microbial community. This community is fundamental to maintaining physiological functions at all ages [11]. The GM composition is unique to each individual and is rapidly established during early childhood, becoming relatively stable by adulthood [12]. The dominant bacterial phyla in the gut include *Firmicutes*, *Bacteroidetes*, *Proteobacteria*, *Actinobacteria*, and *Verrucomicrobia* [13]. This composition can be affected by various environmental factors, such as pH, oxygen levels/redox state, nutrient availability, water activity, and temperature, enabling different microbial populations to flourish and exert diverse activities as they interact with their environment, including the human host [11].

### 2.1. Systemic Effects of the Gut Microbiota

Research has increasingly revealed that the influence of the GM extends beyond the gastrointestinal system, impacting remote organs such as the brain [14,15], lungs [16], heart, kidneys [17], testis [18], and enthesis [19]. The host benefits from a symbiotic relationship wherein it provides a nutrient-rich habitat, while the GM contributes by producing essential metabolic products like vitamins and short-chain fatty acids (SCFAs) which play key roles in the development of metabolic pathways and the maturation of the intestinal immune system [20].

### 2.2. Immunomodulation by Gut Microbiota

During adulthood, the immune system remains continuously stimulated by the GM. This continuous stimulation of the immune system maintains a state of “low-grade physiological inflammation,” enabling rapid defense against pathogens whenever is necessary [21]. GM plays a protective role by competitively metabolizing nutrients essential for the survival of pathogens and producing molecules that inhibit their growth. However, excessive or chronic stimulation, especially of cytotoxic CD8+ T cells, can contribute to immune exhaustion and promote chronic inflammation [22]. Although CD8+ T cells are well known for their anti-tumor and cytotoxic activities, in the presence of an imbalanced GM they can boost damaging inflammation and tumor development [23]. Thus, maintaining a balanced GM is critical to ensure protective immunity while avoiding harmful inflammation.

### 2.3. Role of Short-Chain Fatty Acids (SCFAs)

SCFAs, like acetate, propionate, and butyrate, arise from anaerobic bacterial fermentation of dietary fibers. These metabolites influence host metabolism, immune responses, and inflammation control [24,25]. Butyrate is particularly important as it maintains colonic epithelial barrier integrity, serving as a preferred energy source for colonocytes. It also exhibits anti-cancer properties by augmenting anti-tumor immunity—enhancing CD8+ T cell function—and promoting epigenetic regulation through histone acetylation and inhibition of histone deacetylases [26,27,28]. Activation of G protein-coupled receptors (GPCRs), such as FFAR2 (GPCR43), by SCFAs can trigger signaling cascades that ultimately suppress cancer cell growth and migration [29].

### 2.4. Mechanisms of GM in Carcinogenesis

The GM can influence colorectal carcinogenesis via three major mechanisms: (a) modulating the equilibrium between cell proliferation and apoptosis; (b) regulating immune system activity; and (c) altering the metabolism of dietary components, host-derived molecules, and pharmaceuticals [18]. This bidirectional interaction between microbiota and host immune system is crucial for maintaining homeostasis, and disruption can contribute to tumor initiation and progression.

## 3. Disruption of GM Homeostasis (Dysbiosis) and Gut Barrier Dysfunction

### 3.1. Dysbiosis and Chronic Inflammation

Dysbiosis represents an imbalance in microbial composition and may lead to intestinal epithelial damage and sustained inflammation. Chronic inflammation, characterized by persistent production of reactive oxygen species and pro-inflammatory cytokines, establishes a tumor-supportive microenvironment that facilitates DNA damage and mutagenesis [30,31].

One hypothesis about the link between dysbiosis and barrier dysfunction proposes that both endogenous and exogenous factors contribute to gut barrier impairment. Subclinical mucosal abnormalities, particularly in individuals with a genetic predisposition, may subsequently encourage the growth of opportunistic microbes, promoting the expansion of opportunistic pathogens with increased virulence. These opportunistic microbes can further exacerbate morphological and functional alterations, resulting in pathological consequences like chronic inflammation and clinical symptoms in the host [32].

### 3.2. Macrophage-Mediated Chromosomal Instability

Novel insights have identified a new hypothesis that links dysbiosis and barrier dysfunction. This theory highlights the critical role of commensal gut bacteria in polarizing colonic macrophages toward a pro-inflammatory phenotype, thereby linking the microbiota to cyclooxygenase-2 (COX-2)-mediated inflammation and colorectal carcinogenesis. This macrophage activation triggers a phenomenon termed the “macrophage-induced bystander effect,” whereby activated macrophages release reactive oxygen and nitrogen species, along with pro-inflammatory cytokines, which inflict collateral DNA damage on adjacent epithelial cells. Such damage manifests as chromosomal instability—including double-strand breaks and aneuploidy—thereby promoting mutagenesis and malignant transformation. The upregulation of COX-2 further amplifies this inflammatory milieu, sustaining a feedback loop that perpetuates tissue injury and tumor progression. This mechanism elucidates how a symbiotic microbial ecosystem under dysregulated immune conditions can indirectly drive oncogenic changes through immune-mediated genomic insult, unveiling an intricate interplay between the gut microbiota, host immunity, and colorectal cancer pathogenesis [31,33].

## 4. Key Microorganisms Implicated in Colorectal Cancer

CRC patients exhibit an altered GM profile compared to healthy individuals [34], with enrichment of bacteria typically found in the oral cavity [35], as well as opportunistic and pathogenic species such as *Fusobacterium nucleatum* (FN), colibactin-producing *Escherichia coli*, and enterotoxigenic *Bacteroides fragilis* (ETBF) [34,36]. These microbial shifts contribute causally to inflammation, epithelial injury, and cancer development.

### 4.1. Fusobacterium nucleatum (FN)

*Fusobacterium nucleatum* is a Gram-negative anaerobic bacterium regularly overrepresented in CRC tissues and is implicated in tumor initiation and progression through immunomodulation. Tran et al. identified FN as the most consistently observed marker of excessive colonization in the colon [37]. FN activates the E-cadherin/β-catenin pathway and Toll-like receptor 4 (TLR4) signaling, leading to increased tumor cell proliferation [36,38]. Its membrane protein Fibroblast Activation Protein-2 (FAP2) facilitates adherence to CRC cells by binding to D-galactose-(1-3)-N-acetyl-D-galactosamine (Gal-GalNAc), a carbohydrate overexpressed in CRC tissue, and interacts with immune cell inhibitory receptor immunoreceptor tyrosine-based inhibitory motif (ITIM) on macrophages, dendritic cells, natural killer cells, and various T cell subsets—mediating immune suppression and promoting tumor immune evasion [39]. FN’s adhesin (FadA) promotes bacterial adhesion and inflammation, stimulating CRC cell growth. Notably, anti-FadAc IgA antibodies are elevated in CRC patients, especially those with proximal colon tumors, suggesting diagnostic utility [40,41].

### 4.2. Escherichia coli

Certain strains of *E. coli*, a bacterium commonly found in the human GM, can contribute to the development of CRC. Specifically, phylogroups B2 and D are often linked to intestinal diseases because they produce bacteriocins such as colibactin, which may increase the risk of cancer [42]. The genotoxin can infiltrate colonic cell membranes and translocate to the nucleus. Once there, it induces DNA double-strand breaks, arrests the cell cycle, and interferes with DNA repair mechanisms. Consequently, this process leads to chromosomal aberrations, ultimately contributing to carcinogenesis [43].

Rashid et al. suggest that *E. coli*-produced aglycosylated antibodies (Abs) could be used for therapies targeting CRC. *E. coli*-produced aglycosylated Abs offer advantages such as homogeneity, cost-effectiveness, simplified downstream processing, and suitability for disease indications where effector functions are unnecessary or unwanted. Given their ability to be engineered for specific effector functions and their demonstrated efficacy in other clinical areas, these Abs may offer new opportunities for treating CRC [44].

### 4.3. Enterotoxigenic Bacteroides fragilis (ETBF)

ETBF secretes a metalloprotease toxin (*B. fragilis* toxin, bft) that disrupts epithelial E-cadherin junctions, induces epithelial–mesenchymal transition (EMT), and activates pro-inflammatory cascades, including the Wnt and MAPK pathways, via an interleukin-17A (IL-17A)-dependent mechanism. These actions favor chronic intestinal inflammation and promote tumorigenesis [45,46].

A pivotal investigation within a large European prospective cohort was conducted by Butt et al., who performed a case–control study assessing whether antibody (Ab) reactivity to specific *Escherichia coli* proteins and enterotoxigenic *Bacteroides fragilis* (ETBF) toxin was associated with an increased risk of colorectal cancer (CRC). Remarkably, this study represents the first to directly evaluate serological markers of both *E. coli* and ETBF in relation to CRC susceptibility. The findings revealed that individuals exhibiting IgG dual-positivity to *E. coli* and ETBF had a significantly greater likelihood of developing CRC, with the association being especially strong for tumors arising in the proximal colon. Moreover, Abs against *E. coli* proteins related to biofilm formation and the genotoxin colibactin, as well as Abs targeting the ETBF toxin, were independently linked with an elevated risk of CRC. These serological responses underscore the potential role of specific microbiota-driven immune reactions in promoting colorectal carcinogenesis and highlight promising avenues for future biomarker discovery and risk stratification in CRC [47].

### 4.4. Additional Microbes of Interest

*Enterococcus faecalis* (Efa), an early colonizer of the infant gut, contributes to CRC risk via reactive oxygen species generation and extracellular superoxide production, promoting DNA damage and genomic instability. Efa also stimulates macrophage matrix metalloprotease-9 (MMP-9), which compromises intestinal barrier integrity and fuels inflammation [36,48]. Increased fecal abundance of oral and periodontal pathogens such as *Haemophilus*, *Actinomyces*, and *Porphyromonas* in CRC patients suggests these microbes may contribute to intestinal dysbiosis and carcinogenesis [49].

## 5. Bacteriophages: Emerging Modulators of the Gut Microbiome

Bacteriophages (phages) are the most plentiful biological entities on Earth, with an estimated population exceeding 10^31^ particles. They colonize multiple human body sites, notably the gastrointestinal tract, where their abundance reaches up to 10^8^ particles per milliliter in fecal filtrate. Phages exclusively infect bacteria, modulating bacterial population dynamics and facilitating horizontal gene transfer, especially during inflammatory states [50,51,52].

### 5.1. Dynamics Across Lifespan and Impact of Diet

Although the healthy core phageome in adults is well-characterized, its composition varies considerably throughout an individual’s lifespan. Phage populations in infants are highly dynamic, reflecting the developing bacterial microbiome. Dietary patterns, especially Western diets rich in fats and low in fiber, negatively influence phage diversity and function, mirroring the adverse effects on bacterial communities linked to metabolic and inflammatory diseases [53,54].

### 5.2. Synthetic Phage Engineering and CRISPR-Cas Applications

Recent advances in synthetic biology have enabled the design and engineering of customized bacteriophages with enhanced therapeutic capabilities. These engineered phages can feature extended host ranges to target a broader spectrum of bacterial strains and improved biofilm degradation capacities critical for disrupting resilient microbial communities, elimination of lysogenic cycles to favor strictly lytic activity, and the incorporation of payload genes to deliver additional functionalities. A prominent example of payload engineering involves the integration of the clustered, regularly interspaced short palindromic repeats (CRISPR) and Cas RNA-guided nuclease (CRISPR-Cas) system into phages. Such engineered phages serve as delivery vehicles for programmable, sequence-specific antimicrobials like Cas nucleases, which selectively cleave DNA sequences in bacterial hosts. This targeted cleavage allows bacteria to be re-sensitized to antibiotics by disrupting genes on chromosomes or plasmids that encode virulence factors or antibiotic resistance. In colorectal cancer treatment, this approach aims at modulating the gut microbiota by eliminating harmful bacteria, reducing microbial-driven inflammation, and enhancing therapeutic outcomes [55,56].

Phage therapy has been employed in combating bacterial infections since their discovery and is currently gaining prominence across fields ranging from dentistry to medical microbiology [57]. The growing interest in phage therapy strengthens the rationale for further investigation of phages as modulatory agents in CRC.

### 5.3. Phage Therapy and Immunomodulatory Properties

While bacteriophages are generally considered safe for human health due to their specificity for bacterial hosts and lack of tropism for eukaryotic cells, some phages can interact with the host immune system. For example, the M13 bacteriophage is highly immunogenic, owing to its single-stranded DNA, which can activate various innate immune receptors in immune cells. This immunogenicity enables M13 phages to elicit both humoral and cellular immune responses. As a result, engineered M13 phages displaying tumor-specific antigens represent a promising and cost-effective platform for cancer immunotherapy, particularly aimed at modulating the tumor microenvironment (TME) [58].

Studies investigating the gut virome in colorectal cancer (CRC) patients have revealed distinctive viral signatures associated with disease status. These virome alterations largely stem from shifts in bacteriophage populations, especially enrichment of phages belonging to families such as *Siphoviridae*, *Myoviridae*, *Podoviridae*, *Drexlerviridae*, and *Inoviridae*, which appear to drive this CRC-specific virome profile [54].

Furthermore, Luo et al. examined the relationship between *Helicobacter pylori* infection and CRC, focusing on the gut viral communities. Their findings identified predominant phage classes including *Caudoviricetes*, *Malgrandaviricetes*, and *Faserviricetes*. Notably, most of these phages were temperate, capable of either lysogenic replication as prophages integrated into bacterial genomes or lytic replication producing new phage particles. In the murine gut, a high abundance of lysogens—bacteria harboring prophages—was observed, indicating that inflammatory stimuli may induce prophage activation via oxidative stress and the bacterial SOS response, potentially influencing bacterial population dynamics and CRC pathogenesis [59].

Bacteriophages are garnering growing attention for their potential involvement in CRC carcinogenesis or progression. This is attributed to their capacity to influence the GM and the immune system (Table 1). The utilization of phage-guided nanotechnology to modulate the GM presents a promising avenue for developing innovative approaches in the treatment of CRC [60,61].

**Table 1 ijms-26-07837-t001:** Phage therapies in CRC. MDSCs, myeloid-derived suppressor cells; APCs, antigen-presenting cells; FOLFIRI, irinotecan–fluorouracil–folinic acid; CEA, carcinoembryonic antigen; pRNA, packaging RNA.

In Vitro Studies				
Sample Type	Phage Modulation	Results	Novelty	References
BALB/c and C57BL/6 mice treated with CT26-CEA cells and MC38-CEA cells	Both local and systemic administration of M13 phage against CEA, an over-expressed tumor-associated antigen in CRC.	A significant decrease in tumor growth and extended survival in CRC mice models, as compared to the control group. The infiltration of macrophages and neutrophils into the tumor. The maturation of dendritic cells in the lymph nodes draining the tumor site.	The antitumor impact of M13-CEA is mediated through CD8+ T cells.	Murgas et al., [58]
NIH3T3 and HT-29 CRC cells lines	EGFR-targeted phage lambda (EGF-λ).	EGF-λ interfered with the initial formation and progression of cancer tissues. Moreover, these phages may be able to disturb the formation of metastasis.	λ phages exhibit the ability to traverse fibroblast and ECM layers and accumulate in CRC tissues without presenting cytotoxic effects.	Huh et al., [62]
Primary tumor cells (HCT116, SW480) and metastatic (SW620) CRC cells lines	The hybrid self-assembling nanotubes composed of the tail sheath protein gp053 derived from the phage vB_EcoM_FV3 and a label SNAP-tag protein, obtaining 053SNAP nanotubes.	HCT116 and SW480 efficiently accumulated nanotubes, in contrast with a low uptake in SW620 cells. Internalization into cancer cells escalated over time and did not result in direct cytotoxicity effects. Peritoneal macrophages actively took up 053SNAP, potentially posing a challenge to the utilization of these nanocarriers.	The self-assembling nanotubes generated from phage sheath proteins offer a promising foundation for the future development of innovative nanocarriers.	Gabrielaitis et al., [63]
CRC metastatic cells in liver, lung and lymph node	RNA nanoparticles, originating from the three-way junction of the phage phi29 motor pRNA, were employed to specifically target metastatic CRC cells.	The RNA NPs exhibited a tendency to home in on metastatic tumors without accumulating in the neighboring normal organ tissues.	pRNA-based NPs possess both specificity and advantageous pharmacokinetic characteristics.	Rychahou et al., [64]
HCT116 Human Colon Cancer Cell line co-cultured with *E. faecalis*	Phage EFA1 against EF.	Disrupted EF biofilms within just 2 h of treatment. Altered the growth-stimulatory effects of EF. Suppressed cell proliferation. Increased ROS production.	This study investigates the initial exploration of the impact of Enterococcus bacteriophages on colon cancer cells.	Kabwe et al., [65]
Orthotopic CRC mice and fecal samples of patients with CRC	Encapsulated irinotecan (a primary drug against CRC) within dextran nanoparticles were covalently linked to isolated phages from human saliva that selectively lyse FN.	Inhibition of pro-tumoral bacteria (FN). Promotion of anti-tumoral bacteria (*Clostridium butyricum*) that increase colonic SCFA. Elevated production of anti-tumoral butyrate.	Increase in the expression of anti-autophagy genes (Tspo, Poldip2, and Akt11), along with a decrease in the expression of pro-autophagy genes (Lrrk2, Adrb2, Zc3h12a, Dram1, and Dram2).	Zheng et al., [61]
Stool samples from FN-colonized CRC and orthotopic CT26 murine models	FN binding M13 phage linked to silver nanoparticles (M13@Ag).	Both in vivo and in vitro studies, M13@Ag demonstrated the ability to effectively remove FN from the gut and decrease the amplification of MDSCs in the tumor site, and to activate APCs, contributing to the stimulation of the host immune system.	The combination of M13@Ag with chemotherapy (FOLFIRI) and immune checkpoint inhibitors (α-PD1) resulted in a delayed tumorigenesis and significantly increased the lifespan of mice.	Dong et al., [66]
HTC116 human colon adenocarcinoma and mouse MC38 colon cancer cells; mice with MC38 tumors induced	CEA targeting the expression of E gene (pCEA-E) specifically to colon cancer cells.	Substantial inhibition of cell growth in both human and mice colon cancer cells and a reduction in tumor volume.	The E gene expression presents antitumoral activity: formation of a toxic transmembrane pore, mitochondrial damage, and caspase protein expression.	Rama et al., [67]
CRC Mouse models	*B. fragilis*-targeting phage VA7.	Selective targeting of B. fragilis by phage VA7 restores chemosensitivity in CRC in mice.	A promising adjunct therapy for chemoresistant tumors.	Ding et al., [68]

## 6. Antibiotics and Their Complex Relationship with CRC

In recent years, the critical role of the gut microbiota in colorectal cancer development and therapy has gained substantial recognition. Extensive research has examined antibiotic treatments as a method to directly eliminate or inhibit intestinal microbiota implicated in CRC. However, prolonged and unregulated antibiotic use not only disrupts the normal GM composition but also compromises the host’s natural defenses against colonization and overgrowth of pathogenic bacteria, potentially exacerbating dysbiosis [69,70].

A meta-analysis by Weng et al. demonstrated a correlation between antibiotic exposure and increased CRC risk. Their subgroup analysis revealed a significant elevation in colon cancer risk associated with antibiotics, whereas this association was not evident for rectal cancer. Notably, use of penicillins, cephalosporins, and anti-aerobic and anti-anaerobic antibiotics were linked to higher CRC risk. Conversely, no significant associations were observed with quinolones, macrolides, sulfonamides, tetracyclines, or nitrofurans. The role of metronidazole remains contentious, with some studies suggesting it increases CRC risk [71] while others report protective effects [72,73].

Lu et al.’s comprehensive nationwide Swedish cohort study further reinforced the association between antibiotics and increased risk of proximal colon cancer. Intriguingly, an inverse relationship was noted for rectal cancer in women. Their findings indicated that even minimal antibiotic exposure elevates proximal colon cancer risk, supporting calls for prudent antibiotic use in clinical practice [74].

In the context of cancer immunotherapy, Routy et al. revealed that antibiotic-induced disruption of the GM negatively impacts the efficacy of immune checkpoint inhibitors (ICIs). Advanced cancer patients receiving antibiotics near the time of ICI therapy experienced diminished clinical benefits, highlighting the microbiome’s critical role in modulating treatment responses. Importantly, administration of *Akkermansia muciniphila* following fecal microbiota transplantation from nonresponders restored PD-1 blockade efficacy in a manner dependent on interleukin-12 and enhanced recruitment of tumor-infiltrating T lymphocytes in murine models [75]. Furthermore, nonresponders to ICIs typically exhibit a GM dominated by *Proteobacteria* genera (e.g., *Escherichia coli*, *Shigella*, *Klebsiella* spp.), which are commonly linked to prior antibiotic exposure. In addition, elevated levels of *Fusobacterium*, members of the *Porphyromonadaceae* family, and *Actinobacteria* species such as *Eggerthella lenta* and *Atopobium parvulum* characterize nonresponders, although their precise roles in immunosuppression remain to be fully elucidated [76,77].

The microbiome not only influences tumorigenesis but also plays a central role in the host response to cancer (Table 2). The utilization of antibiotics to manipulate TME as a therapeutic strategy against CRC remains a topic of debate [73,78].

## 7. Probiotics: Potential Adjuncts in CRC Management

Probiotics are defined as “live microorganisms that confer health benefits on the host when administered in adequate amounts” [91,92]. Their beneficial effects are often species- and strain-specific and primarily involve restoration of a balanced gut microbiota (GM). These effects include reversal of dysbiosis, prevention of pathogenic bacterial infections and their mucosal adhesion, and reinforcement of intestinal barrier integrity [91,92].

In the context of colorectal cancer, probiotics have demonstrated potential to reduce inflammatory markers, alleviate gastrointestinal tract inflammation, and decrease the incidence of inflammation-associated carcinogenesis. They also contribute to cancer prevention by binding and metabolizing harmful carcinogens. Additionally, probiotics promote the production of anti-cancer metabolites such as short-chain fatty acids (SCFAs), which exert anti-inflammatory and antiproliferative effects [93]. Notably, probiotics may act synergistically with immune checkpoint inhibitors (ICIs) to enhance anti-tumor immune responses, suggesting an important complementary role in cancer therapy [94].

The most studied probiotic genera in CRC are *Lactobacillus* and *Bifidobacterium*, both natural residents of the human digestive tract capable of fermenting sugars and producing lactic acid. Lactobacillus species confer diverse benefits, including antibacterial, anti-inflammatory, antioxidant, and immunomodulatory effects. They generate antioxidants and antiangiogenic factors that reduce DNA damage, inflammation, and tumor size. Moreover, they inhibit the expression of tumor-specific proteins, polyamines, and procarcinogenic enzymes, collectively contributing to CRC prevention and therapy [10,95].

Combinations of probiotics also show promise. A synbiotic cocktail comprising *Bifidobacterium bifidum*, *B. longum*, *Lactobacillus* acidophilus, and *L. plantarum*, combined with prebiotics such as fructo-oligosaccharides, resistant dextrin, isomalto-oligosaccharides, and stachyose, exhibited antiproliferative effects on mouse colon cancer cells. This formulation also decreased metastatic properties, including migration and invasion, primarily via a T-cell-mediated immune response characterized by increased CD8+ T cell infiltration [96,97]. Clinical studies have reported that probiotic strains of *Lactobacillus* and *Bifidobacterium* are safe and demonstrate anti-inflammatory properties when administered postoperatively to CRC patients [98].

Specifically, *Lactobacillus acidophilus* shows potential as an adjunct therapy in rectal cancer. It reduces inflammation by upregulating the cylindromatosis (CYLD) protein, which negatively regulates the nuclear factor kappa-light-chain-enhancer of activated B cells (NF-κB) signaling pathway in tumor tissues [99]. Additionally, this probiotic modulates gene regulatory networks involved in cancer behavior, decreasing oncogenic microRNAs (miRs) and long non-coding RNAs (lncRNAs), while enhancing tumor-suppressive miRs and lncRNAs. These multifaceted effects suggest that *L. acidophilus* supplementation could improve prognosis by targeting both inflammation and genetic regulation in rectal cancer management [100].

Emerging candidates for next-generation probiotics in CRC include *Clostridium butyricum*, *Bacteroides fragilis*, *Faecalibacterium prausnitzii*, and *Akkermansia muciniphila* [101]. *C. butyricum* is a major butyrate-producing symbiont that reinforces gut barrier integrity and maintains immune homeostasis by balancing pro- and anti-inflammatory responses, promoting regulatory T cell (Treg) induction [102]. It downregulates overexpression of Methyltransferase-like 3 (METTL3), an enzyme implicated in promoting epithelial–mesenchymal transition and vasculogenic mimicry, thereby restricting CRC cell proliferation and invasion [103].

*F. prausnitzii* is another predominant butyrate producer in the GM. Dikeocha et al. demonstrated that the cell-free supernatant of *F. prausnitzii* inhibited proliferation of HCT116 CRC cells in a time-dependent manner [104]. Moreover, intragastric administration of *F. prausnitzii* combined with intraperitoneal injection of tyrosol (an olive oil metabolite) enhanced CD8+ T cell infiltration into tumor tissues in vivo, indicating immunomodulatory potential within the TME [105].

The role of *Akkermansia muciniphila* in CRC remains controversial. While some studies report elevated abundance of *A. muciniphila* in CRC patients, other findings associate its depletion with more severe clinical symptoms [106]. Derosa et al. proposed *A. muciniphila* as a potential biomarker for predicting clinical responses to PD-1 blockade in advanced non-small-cell lung cancer. Responders to PD-1 inhibitors exhibited significantly higher abundance of *A. muciniphila* compared to non-responders, along with a bacterial community associated with healthy immunogenic status—characterized by families *Ruminococcaceae* (including *F. prausnitzii* and *Ruminococcus lactaris)* and *Lachnospiraceae* (e.g., *Dorea formicigenerans*, *D. longicatena*, *Eubacterium hallii*, *E. rectale*, *Roseburia intestinalis*, *R. faecis)*, and species such as *Bifidobacterium adolescentis* and *Intestinimonas butyriciproducens* [107]. This suggests that *A. muciniphila* may influence immunotherapy outcomes in CRC and other cancers. Its presence has also been linked to improved therapeutic efficacy in prostate cancer via androgen deprivation therapy and in renal cancer by enhancing immune checkpoint blockade effectiveness [108,109].

Innovative probiotic therapies have been explored, exemplified by a trial from Gurbatri et al. utilizing bioengineered *Escherichia coli* for CRC detection and treatment. When orally administered, this probiotic selectively colonizes colorectal adenomas in genetic and orthotopic CRC models, producing salicylate detectable in urine—a potential non-invasive early detection marker. These engineered strains also secrete granulocyte-macrophage colony-stimulating factors (GM-CSFs) and nanobodies targeting CTLA-4 and PD-L1 checkpoint proteins, achieving a 50% reduction in adenoma burden, underscoring exciting therapeutic possibilities [110].

We identified seven studies exploring the clinical application of probiotics in CRC patients, focusing on their preoperative and postoperative benefits, ability to prevent chemotherapy side effects, and capacity to improve overall outcomes (Table 3). The future of probiotics in the context of CRC shows promising potential. Ongoing research and clinical studies are presumptive to clarify the intricate relationship between GM, probiotics, and CRC.

**Table 3 ijms-26-07837-t003:** The use of probiotics in clinical trials in CRC patients. QoL, quality of life.

Study Design/ Sample Size	Probiotics	Results	Clinical Importance/Novelty	References
Randomized, single-blind, placebo-controlled prospective study/100 patients	*B. infants*, *L. acidophilus*, *E. faecalis*, *and B. cereus*	↓ severity of chemotherapy-induced gastrointestinal complications; ↑ levels of SCFA; ↑ gut microbiota homeostasis.	Probiotics have the potential to be used as complementary approaches for chemoprevention.	Huang et al., [111]
Randomized, double-blind, placebo-controlled trial/52 patients	*L. acidophilus*, *L. lactis*, *L. casei*, *B. longum*, *B. bifidum* and *B. infantis*	↓ IL-6, IL-10, IL-12, IL-17A, IL-17C, IL-22 and TNF-α. No difference in the levels of IFN-γ.	These strains of *Lactobacillus* and *Bifidobacteria* were shown to be safe and to have anti-inflammatory properties when consumed after surgery.	Zaharuddin et al., [98]
Randomized, double-blind, placebo-controlled trial/66 patients	*L. rhamnosus*, *L. acidophilus*	↑ QoL with improved mental health status; ↓ cancer-related fatigue.	Probiotics could improve both physical and psychological health.	Lee et al., [112]
Randomized controlled trial/110 patients	*L. acidophilus*	↓ inflammation by influencing CYLD protein.	The first study to investigate the link between *L. acidophilus*, CYLD expression and NF-κB/TNF-α signaling.	Zamani et al., [99]
Double-blind randomized trial/140 patients	*L. acidophilus*, *L. casei*, *L. lactis*, *B. bifidum*, *B. longum*, *B. infantis* + omega-3 fatty acid	↑ QoL; ↓ IL-6; ↓ chemotherapy-related side effects.	This combination of probiotics and omega-3 might act as a synergic adjuvant to chemotherapy.	Golkhalkhali et al., [113]
Randomized prospective study/78 patients	*L. acidophilus*, *L. casei*, *L. plantarum*, *L. rhamnosus*, *B. lactis*, *B. bifidum*, *B. breve*, *Streptococcus thermophilus*	↓ postoperative complications and hospitalization; ↓ post-surgery mortality within the first six months.	Incorporating probiotics into the postoperative care treatment for CRC patients could lead to improved outcomes.	Bajramagic et al., [114]
Randomized, prospective, double-blind, placebo-controlled study/73 patients	*L. acidophilus*, *L. rhamnosus*, *L. casei*, *B. lactis * + Fructose Oligosaccharide	Used preoperatively. ↓ inflammatory state; ↓ morbidity and hospitalization; ↓ duration of antibiotic usage; ↑ bowel functions.	Modulating the microbiota during the preoperative period in CRC patients could influence the incidence of postoperative complications.	Polakowski et al., [115]

↓ = decrease; ↑ = increase.

## 8. Conclusions

The intricate interplay between the GM, bacteriophages, antibiotics, and probiotics holds profound implications for CRC development, progression, and treatment.

As reviewed, dysbiosis of the GM contributes to CRC pathogenesis by promoting chronic inflammation, disrupting the intestinal barrier, and facilitating genotoxic effects through specific bacterial species such as *Fusobacterium nucleatum*, *Escherichia coli*, and *Bacteroides fragilis*.

Bacteriophages emerge as promising agents in CRC management due to their ability to selectively infect and lyse specific bacterial populations. Unlike broad-spectrum antibiotics, phages offer targeted modulation of pathogenic bacteria implicated in CRC, such as *F. nucleatum*. However, given their potential to inadvertently impact commensal, beneficial microbes, careful characterization and engineering of phage therapy are imperative to preserve microbial balance and avoid worsened dysbiosis.

Antibiotics exert multifaceted effects on CRC cells, including inhibition of tumor proliferation, inflammation, and angiogenesis. Nevertheless, their extensive and often indiscriminate use raises significant concerns related to antimicrobial resistance and collateral damage to the GM. This disruption may exacerbate dysbiosis, diminish beneficial microbial metabolites like short-chain fatty acids, and ultimately impair host immune responses against tumors. The association between antibiotic exposure and increased CRC risk, particularly for specific antibiotic classes and proximal colon cancers, necessitates judicious prescribing practices.

Probiotics offer a beneficial counterbalance by restoring GM equilibrium, enhancing mucosal barrier integrity, and modulating immune responses. Probiotic strains such as *Lactobacillus acidophilus*, *Bifidobacterium species*, and next-generation candidates like *Clostridium butyricum* and *Faecalibacterium prausnitzii* contribute to anti-inflammatory effects and may potentiate anti-tumor immunity, including synergistic effects with immune checkpoint inhibitors. Additionally, bioengineered probiotics hold promise for innovative diagnostics and therapeutic delivery tailored to CRC.

The combined use of bacteriophages, antibiotics, and probiotics represents a synergistic potential to modulate the GM and TME with enhanced precision. Future approaches integrating these modalities could revolutionize CRC management by eradicating harmful bacteria while supporting protective microbiota and host immunity.

Nonetheless, the complex interactions within the GM ecosystem and the TME underscore the need for comprehensive research. Robust clinical trials and mechanistic studies are essential to elucidate optimal phage therapies, antibiotic use strategies, and probiotic formulations, ensuring safety and maximizing therapeutic efficacy. Ultimately, personalized microbiome-centered interventions based on patient-specific microbial and immune profiles may emerge as powerful tools in CRC prevention and treatment.

## Figures and Tables

**Figure 1 ijms-26-07837-f001:**
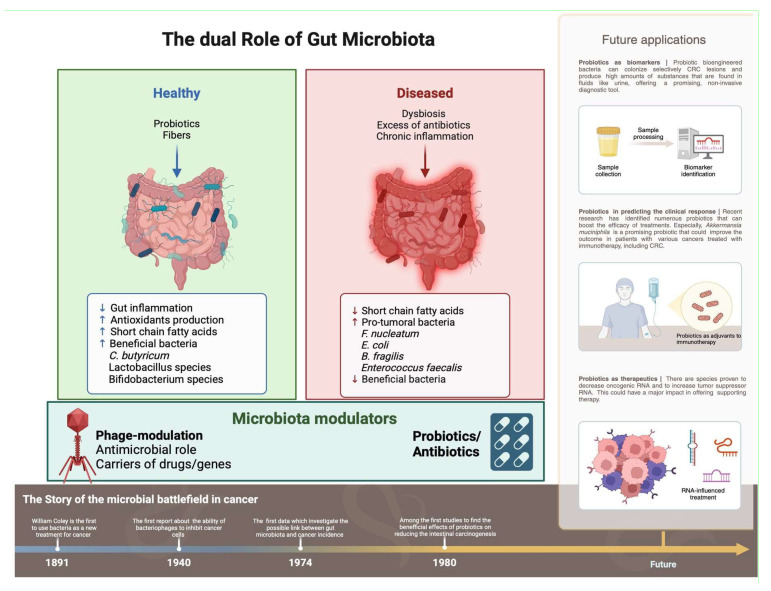
The systematized impact of GM in CRC. ↓ = decrease; ↑ = increase; (BioRender.com).

**Table 2 ijms-26-07837-t002:** Antibiotics used in pre-clinical trials investigating the treatment of CRC.

Antibiotics	Sample Type	Results	Novelty	References
Metronidazole	Human primary CRC mice xenografts	↓ FN load; ↓ Cancer cell proliferation; ↓ Tumor growth.	Further antimicrobial interventions are needed for FN-associated CRC patients.	Bullman et al., [72]
Monensin	Human CRC cell lines RKO and HCT-116	Induction of G1 arrest and apoptosis. ⊝ cell migration and proliferation. ⊝ IGF1R expression.	This study highlights the importance of IGF1R signaling pathway.	Zhou et al., [79]
	HCT-116, HT29	Modulation of TLR4/IRF3 signaling pathway. Anti-inflammatory effect.	The first evidence establishing a connection between monensin and TLRs.	Seçme et al., [80]
Rapamycin liposomes and 5-Fluorouracil	Azoxymethane/dextran sulfate sodium induced CRC mouse model	↓ tumorigenesis; ↓ number of tumors. Anti-angiogenesis effect. ⊝ proliferation, migration, and tube formation of HUVECs.	This combination may represent an important starting point for more investigations.	Liu et al., [81]
Salinomycin	Primary tumor-initiating cells derived from CRC patients	Cell death. ⊝ proliferation; ⊝ respiratory chain complex II. ↓ ATP production; ↓ expression of SOD1; ↑ ROS and dysfunctional mitochondria.	These studies underline the need for additional research about analogs of salinomycin or combinations with this challenging toxic drug.	Klose et al., [82]
Semi-synthetic salinomycin	SW480 and SW620 human CRC cell lines	↑ effects at lower concentrations compared to salinomycin.		Klose et al., [83]
Salinomycin and sulforaphane	Human CRC Caco-2 and CX-1 cell lines	Antiproliferative and proapoptotic effects. ↑ p53 expression; ↓ Bcl-2 expression.		Liu et al., [84]
Nigericin	SW620 and KM12 human CRC cell lines	↑ tumor apoptosis; ↓ tumor cell proliferation; ↓ β-catenin and TCF-1 expressions; ↓ cyclinD1, Survivin, Axin2, MMP-7/-9, and c-Myc.	Nigericin was efficient in the suppression of both tumor growth and metastasis in CRC cells.	Liu et al., [85]
	HCT-116 cell lines	Apoptosis and autophagy inhibition of GSK-3β and JAK3 kinases.		Garcia-Princival et al., [86]
Tigecycline	HCT-116 cell lines Colitis-associated CRC murine model	↓ tumorigenesis; ↓ proliferation (⊝ STAT3 activation and Wnt/β-catenin pathway); ↓ cancer-associated inflammation; ↑ cytotoxic T lymphocytes activity; ↑ apoptosis (↑ CASP7); ↑ protective anti-tumor bacterial genera and species (*Akkermansia*, *Parabacteroides distasonis*).	This study demonstrates the potential benefits of using tigecycline and brings up a very promising drug against CRC.	Ruiz-Malagón et al., [87]
Minocycline	SW480 and SW620 cells	↓ metastasis; ⊝ LYN activity; ⊝ STAT3 signaling; ⊝ epithelial–mesenchymal transition.	Via direct binding to LYN, an enzyme correlated with increased activity and metastasis development in CRC, minocycline may serve as an effective treatment.	Yang et al., [88]
Antimycin A	HCT-116 cells	↑ p53 and CASP-9 gene expression; ↓ MAPK, PARP, and NF-kB genes.	The first study to investigate the antimycin A apoptotic pathways in CRC cells.	Kabir et al., [89]
Clarithromycin	HCT-116 cells	↑ 5-Fluorouracil cytotoxic effect; ↑ apoptosis and cytosolic autophagosomes; ⊝ the assembly of a macromolecular structure between hERG1 and p85 component of PI3K.	As hERG1 is increased in aggressive primary CRC, clarithromycin or other drugs aiming hERG1 need further investigations in clinical trials.	Petroni et al., [90]

↓ = decrease; ↑ = increase; **⊝** = inhibition.

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
