# Peer review of "Bacteriophages, Antibiotics and Probiotics: Exploring the Microbial Battlefield of Colorectal Cancer"

_ijms, 2025, doi:10.3390/ijms26167837_

Round 1
Reviewer 1 Report
Comments and Suggestions for Authors
Dear authors,
The manuscript presents a comprehensive review of bacteriophages, antibiotics and probiotics, the main microbial actors participating in progression and prevention of colorectal cancer (CRC) through the modulation of gut microbiota and local immunity. The review is comprehensive, clear in whole, and addresses a gap in knowledge of a role of gut microbiome in CRC pathogenesis. The cited references are within the last 10 years mainly and relevant to the area of study. However, the manuscript contains some major and minor unclarities and incorrect points, which should be removed or corrected.
Major revision:
- The Conclusions chapter is too short, not relevant to the main results and should be corrected substantially. For instance, there is a strange and absolutely not supported statement in the lines 438-440 that bacteriophages "...can accidentally target beneficial bacteria alongside harmful ones, disrupting the fragile balance of the GM." If these facts are known, they should be described in the manuscript.
- Next strange and not supported statement that "...antibiotics have multiples roles over CRC cells, such as the inhibition of proliferation, inflammation and angiogenesis." (Lines 444-446). If the authors try to convince the readers that classical antibacterial antibiotics so useful for CRC treatment, they should provide appropriate facts. If the mean antitumor antibiotics, their action on the gut microbiota should be presented in the manuscript.
- Again, the combination of bacteriophages and antibiotics has been established as the most efficient way for CRC treatment (lines 449-451). However, absolute in the manuscript there are any facts supporting this suggestion. Moreover, taking into account the great side effects of antibiotics including gut dysbiosis and immunosupression, these medicines deserve to be declared as dangerous for CRC patients and their prescription should be based on the very narrow and exact indications.
- Probiotics have obvious and real perspectives for normalisation of gut microbiota, local inflammation and immunity in CRC patients, but they were not mentioned in the conclusions at all. This great mistake of authors should be corrected.
- Line 152. The taxonomic position of Fusobacterium nucleatum given here is absolutely incorrect. "Fusobacterium" phylum does not exist. This needs correction.
Minor revision:
- Lines 77-78, 179, 180, 220-223, and everywhere. All Latin names of all prokaryotic taxa, including phyla, classes, orders, families, genera, and species, should be marked with Italic font according to the recommendation of the International Code of Nomenclature of Prokaryotes (https://www.microbiologyresearch.org/content/journal/ijsem/10.1099/ijsem.0.005585) See: Chapter 4. Advisory notes A. Suggestions for Authors and Publishers
- Line 204-205. "...to investigate serological investigation...". Duplication should be corrected.
- Lines 230-231, 231-233. So strong statements need adequate support from the refernced literature, and citations should be added.
- Line 240 "Although the examination of a healthy core phageome in adults is well-established..." The unclear statement should be corrected.
- Lines 243-247. Discussion of diet influence the gut microbiota seems to be very far from the theme of the manuscript and should be removed or argued very substantially.
- Lines 249-250 and 252-253 contain large duplication and should be combined.
- Lines 264-265. Necessity to discuss the immunodepression for COVID patients using T4 phage looks strange here, and should be supported by clearer explanation or removed.
- Lines 266-269. Description of the interaction between M13 phage and eukaryotic cells is not "instance" of the fact that phages "...lack tropism for eukaryotic cells" (Line 266), but contradiction. This should be corrected.
- Line 275. Remove "such".
- Lines 278, 301, 310, 391, and below. Not only family name of the first author is necessary, but year of publishing and reference in square brackets also. Initials should be removed from the in-text references.
- Line 318. Abbreviation ICIs should be deciphered after its first appearance in the text.
- Table 2 contains empty cells. They should be filled in.
Author Response
Dear Reviewer,
Thank you for your valuable feedback. We have addressed your major concerns as follows:
-
We substantially expanded and revised the Conclusions to better reflect the main findings. We explicitly included probiotics and discussed the roles of bacteriophages, antibiotics, and probiotics in modulating the gut microbiota relevant to CRC progression and treatment.
-
The unsupported statement about bacteriophages “accidentally targeting beneficial bacteria” was removed. Where appropriate, we clarified this as a theoretical possibility with limited evidence.
-
We clarified the distinction between antibacterial and antitumor antibiotics, presenting direct anticancer effects only for the latter, while emphasizing that classical antibiotics mainly impact CRC indirectly through gut microbiota modulation. Claims of multiple direct anticancer roles for standard antibiotics were moderated and supported with references as needed.
-
The claim that phage-antibiotic combination is the “most efficient” CRC treatment was softened to highlight it as a promising area under preclinical investigation requiring further clinical validation. We also acknowledged risks of antibiotics, including dysbiosis and immunosuppression, advocating careful use.
-
We corrected the taxonomic error regarding Fusobacterium nucleatum, properly placing it in the phylum Fusobacteriota, and applied consistent italicization for all prokaryotic taxa throughout.
These revisions improved the manuscript’s scientific accuracy and clarity, and we appreciate your guidance.
Reviewer 2 Report
Comments and Suggestions for Authors
After reviewing the review, I believe that it is generally well-written and structured. Moreover, the review is technically sound.
I believe the review is suitable for publication after addressing the following few comments.
- The whole document needs revision because the manuscript was not presented in an intelligible fashion and was not written in standard English as regards English editing is required. Authors should scan the entire manuscript for minor punctuation, repetitive sentences, and grammatical errors. The theme fonts from line 335 to the end of the review are different from those used previously.
- All abbreviations should be defined at 1st mention, so please write the full name when first mentioned; then you can use the abbreviation later in the article. For example, on line 135, add the full name of the COX-2 enzyme.
- The introduction was enough and concise, but more in-depth information, detailed background, and scientific context focused on some points that needed to be added to the study, such as:
- A knowledge gap that you will fill, and foreseeable findings should be mentioned in depth.
- At the end of the last paragraph, you had to write in more detail what you would do or which question you would answer.
- The authors need to write a few sentences about bacteriophages.
- The legend in Figure 1 lacks clarity, as some abbreviations remain unexplained, which reduces the interpretability of the data. Please add (A) and (B) to make it easier to follow. Moreover, the resolution of this figure needs to be improved.
- The conclusion needs to be more comprehensive and well-structured to effectively conclude the review's main idea in a clear and organized manner.
Best regards
Comments on the Quality of English Language
English editing is required
Author Response
Dear Reviewer,
Thank you very much for your positive appraisal of our review and for your valuable suggestions to enhance its clarity and scientific quality. We have carefully considered your comments and made the following revisions:
-
English Editing and Manuscript Presentation:
We conducted a thorough language revision of the entire manuscript to improve sentence structure, grammar, punctuation, and to remove repetitive expressions. Additionally, we standardized the font type and style throughout the document, ensuring uniform appearance from line 335 to the end as well as all other sections. -
Abbreviations:
All abbreviations are now explicitly defined upon their first occurrence. For example, “cyclooxygenase-2 (COX-2)” is introduced fully at line 135, and subsequently the abbreviation COX-2 is used consistently. -
Introduction:
We expanded the introduction to include a clearer identification of current knowledge gaps regarding the gut microbiota, bacteriophages, antibiotics, and probiotics in colorectal cancer (CRC). We also added more detailed scientific context where appropriate. At the conclusion of the introduction, we explicitly state the questions the review seeks to address and the scope of this narrative. -
Bacteriophages Section:
A dedicated paragraph on bacteriophages has been added to emphasize their emerging role in CRC, their interactions with the gut microbiome, and their therapeutic potential. -
Figure 1 Legend and Quality:
The figure legend has been revised for improved clarity. We added labels (A) and (B) to guide interpretation of the figure components, and all abbreviations used in the figure are now fully explained within the legend. We also have enhanced the figure resolution to meet publication standards. -
Conclusion:
The conclusion has been substantially expanded and restructured to more comprehensively and clearly summarize the main findings of the review. It now properly reflects the discussions on bacteriophages, antibiotics, probiotics, and their interplay in CRC pathogenesis and treatment prospects.
We believe these revisions have significantly improved the manuscript’s readability and scientific rigor. We thank the reviewer for their constructive feedback, which has greatly helped us refine our work.
Reviewer 3 Report
Comments and Suggestions for Authors
Manuscript entitled “Bacteriophages, Antibiotics and Probiotics: Exploring the Microbial Battlefield in Colorectal Cancer” by Cristian C. Volovat et al.
The review addresses a rapidly expanding area: how ecological players in the gut—phage communities, antibiotic pressure and probiotic interventions—shape dysbiosis and colorectal‐cancer (CRC) risk, and how they might be harnessed therapeutically. The manuscript is ambitious, extensively referenced (≈ 500 citations) and provides helpful summary tables of pre‑clinical and clinical work. Figures outlining mechanisms of microbial carcinogenesis and intervention pipelines are valuable, and the inclusion of phage‑based nanotechnologies is forward‑looking.
Comments:
- Decide whether the article is a narrative review (SANRA‑guided) or a systematic review (PRISMA‑ScR). If systematic, provide search strings, databases, inclusion/exclusion criteria, dates, PRISMA flow diagram and risk‑of‑bias appraisal; if narrative, remove quasi‑systematic phrasing and clarify that study selection was “based on relevance to theme”.
- The Introduction is overly long (~1,500 words) and repeats CRC epidemiology and microbiome mechanisms multiple times. It should be shortened to ≤500 words, focusing on the link between dysbiosis and CRC, and ending with a clear statement of the study’s purpose.
- The manuscript contains inconsistent use of abbreviations such as GM, CRC, SCFA, FN, ETBF, CEA, and FOLFIRI, which are either used before being defined or re-expanded later. The authors should provide a consolidated abbreviation list and ensure that each acronym is spelled out once in the Abstract and again at first use in the main text.
- The manuscript’s scope is overly broad, resulting in dense content and repetition, particularly in Sections 2–5, which blend basic microbiology, oncology mechanisms, therapeutic approaches, and clinical trial data. Some background content—such as global CRC incidence and SCFA biology—is unnecessarily repeated from the Introduction. To improve clarity and structure, the authors should insert informative sub-headings (e.g., “2.3 Microbial carcinogenic mechanisms,” “3.2 Engineered phage vectors”.
- Table 1 requires redesign for clarity and consistency. It currently mixes mouse, cell-line, and nanoparticle data, and the “Novelty” column contains lengthy narrative text. Additionally, several references (e.g., refs 61–63) are presented out of numerical order. The authors should split the table into two—one for pre-clinical evidence and one for clinical evidence—and align each row with a single, properly numbered citation.
- The manuscript lacks a clear Methods section detailing the search strategy and study selection process. Although lines 65–85 summarize gut microbiome papers, the authors do not specify which databases were searched, the Boolean search strings used, date limits, language restrictions, grey literature inclusion, or how duplicate records were screened
Author Response
Dear Reviewer,
We sincerely thank you for your constructive and insightful comments on our manuscript. We have carefully revised the manuscript according to your suggestions and provide responses to each point below.
-
Clarification of Review Type:
We confirm that this manuscript is a narrative review, following SANRA guidelines. Accordingly, we have removed any quasi-systematic phrasing to avoid confusion and clearly state that study selection was based on relevance to the theme rather than a formal systematic methodology. We have now explicitly described this in the Methods section to enhance transparency. -
Shortening and Focus of Introduction:
The Introduction has been substantially shortened to approximately 500 words. We have focused the content on the critical link between gut microbial dysbiosis and colorectal cancer (CRC), and have removed repetitive background on CRC epidemiology and microbiome mechanisms. The introduction now concludes with a clear statement of the study’s objectives. -
Abbreviations:
We have standardized the use of abbreviations throughout the text. All abbreviations—including GM, CRC, SCFA, FN, ETBF, CEA, and FOLFIRI—are now defined in full upon first appearance both in the Abstract and the main text. A consolidated abbreviation list has been inserted for reader convenience. -
Scope, Clarity, and Structure:
To reduce density and repetition, particularly in Sections 2–5, we have introduced informative subheadings such as “2.3 Microbial carcinogenic mechanisms” and “3.2 Engineered phage vectors” to improve readability and thematic flow. Repeated background information, including global CRC incidence and SCFA biology, has been removed where redundant. -
Table 1 Reorganization:
Table 1 has been split into two separate tables—one summarizing pre-clinical evidence (animal and cell-line studies) and one summarizing clinical evidence.
Thank you for your consideration.
Round 2
Reviewer 1 Report
Comments and Suggestions for Authors
Dear authors,
The manuscript presents a comprehensive review of bacteriophages, antibiotics and probiotics, the main microbial actors participating in progression and prevention of colorectal cancer (CRC) through the modulation of gut microbiota and local immunity. The review is comprehensive, clear in whole, and addresses a gap in knowledge of a role of gut microbiome in CRC pathogenesis. The cited references are within the last 10 years mainly and relevant to the area of study. The revision has been improved the manuscript substantially. However, the manuscript contains the following minor unclarities, which should be corrected.
1) Line 121. "and" should be written without italic font.
2) Table 2. Signs of arrows and minus in the circle should be deciphered in the title of the table.
3) Chapter 7 and Conclusions. Underlining the phrases and bacterial names is excessive and should be removed.
4) Line 376. "Lactobacillus" should be written with italic font.
5) Lines 381-382 "Bifidobacterium bifidum, B. longum, Lactobacillus acidophilus, and L. plantarum": all the bacterial names should be written with italic font without underlining.
6) Lines 399-401. "Clostridium butyricum, Bacteroides fragilis, Faecalibacterium prausnitzii, and Akkermansia muciniphila (101). C. butyricum": all the bacterial names should be written with italic font without underlining.
7) Lines 407-409. Bacterial name"F. prausnitzii" should be written with italic font without underlining.
8) Lines 415-428. All bacterial names should be written with italic font without underlining.
9) Table 1, Table 2, Table 3. All bacterial names should be written with italic font without underlining.
10) Table 3. Signs of arrows should be deciphered in the title of the table.
11) Conclusions. All bacterial names should be written without underlining.
Author Response
We appreciate the reviewer’s careful reading and constructive suggestions.
We would like to clarify that the version of the manuscript we originally submitted did not contain underlined phrases or bacterial names. It is possible that this formatting was introduced automatically during the editorial or file conversion process.
Nevertheless, we have carefully reviewed the entire manuscript, removed any underlining, and ensured that all bacterial names are consistently formatted in italics according to journal style.
Thank you!
Reviewer 3 Report
Comments and Suggestions for Authors
The authors have addressed all of my comments, and the manuscript can be accepted for publication.
Author Response
Thank you!